# CLIPDraw: Exploring Text-to-Drawing Synthesis through Language-Image Encoders

**Kevin Frans**
Massachusetts Institute of Technology, Cambridge, MA, USA
Cross Labs, Cross Compass Ltd., Tokyo, Japan
`kvfrans@csail.mit.edu`

**L. B. Soros**
Cross Labs, Cross Compass Ltd., Tokyo, Japan

**Olaf Witkowski**
Cross Labs, Cross Compass Ltd., Tokyo, Japan
Earth-Life Science Institute, Tokyo Institute of Technology, Japan
College of Arts and Sciences, University of Tokyo, Japan

## Abstract

CLIPDraw is an algorithm that synthesizes novel drawings from natural language input. It does not require any additional training; rather, a pre-trained CLIP language-image encoder is used as a metric for maximizing similarity between the given description and a generated drawing. Crucially, CLIPDraw operates over vector strokes rather than pixel images, which biases drawings towards simpler human-recognizable shapes. Results compare CLIPDraw with other synthesis-through-optimization methods, as well as highlight various interesting behaviors of CLIPDraw, such as satisfying ambiguous text in multiple ways, reliably producing drawings in diverse styles, and scaling from simple to complex visual representations as stroke count increases.

## 1 Introduction

When humans hear a description of a scene, it's easy to imagine what it may look like. Conversely, when we construct a mental image, it's easy to then describe that scene. At some level, humans have a deeply coupled representation for textual and visual structures key to understanding our world.

The recent introduction of CLIP (Radford et al., 2021), a dual language-image encoder, is a large step towards unifying textual and visual information. In a CLIP model, both text and images are mapped onto the same representational space, thus enabling the similarity between images and textual descriptions to be measured. When trained on large amounts of data, CLIP representations have been shown to solve a robust range of image-based recognition tasks.

This work presents *CLIPDraw*, an algorithm that synthesizes novel drawings based on natural language input. CLIPDraw does not require any training; rather a pre-trained CLIP model is used as a metric for maximizing similarity between the given description and a generated drawing. Rather than photorealistic images, CLIPDraw aims to synthesize simple drawings that nevertheless match the prompt. Thus, CLIPDraw optimizes a set of vector strokes rather than pixel images, a constraint that biases drawings towards simple human-recognizable shapes.

36th Conference on Neural Information Processing Systems (NeurIPS 2022).

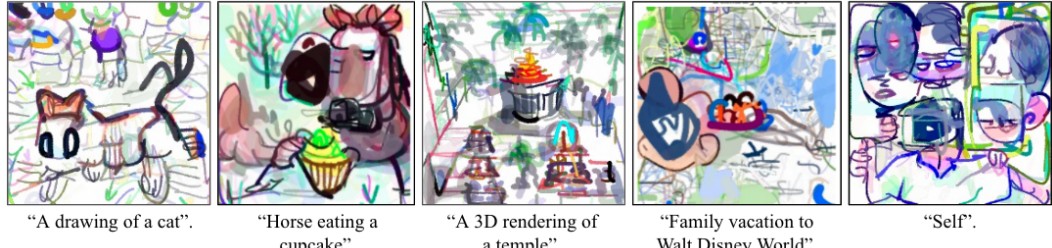

| "A drawing of a cat". | "Horse eating a cupcake". | "A 3D rendering of a temple". | "Family vacation to Walt Disney World". | "Self". |

Figure 1: **Various drawings synthesized by CLIPDraw**, along with the corresponding description prompts used. CLIPDraw synthesizes images from text by performing gradient descent over a set of RGBA Bézier curves, with the goal of minimizing cosine distance between the CLIP encodings of generated images and description prompts. CLIPDraw does not require learning a new model, and can generally synthesize images within a minute on a typical GPU.

The aim of this work is to present CLIPDraw a testbed for exploring language-image relationships and synthesizing AI-assisted artwork, as well as to showcase various nuances of the method. Results compare between CLIPDraw and other optimization-based text-to-image methods, along with highlighting several interesting behaviors:

- By adjusting descriptive adjectives, such as "watercolor" or "3D rendering", CLIPDraw produces drawings of vastly different styles.
- CLIPDraw often matches the description prompt in creative ways, such as writing words from the prompt inside the image itself, or interpreting ambiguous nouns in multiple ways.
- Running CLIPDraw with a low stroke count results in cartoonish drawings, while high stroke counts tend to result in realistic renderings.
- By giving CLIPDraw abstract prompts, such as "happiness" or "self", we can examine what visual concepts the CLIP model associates with them.
- CLIPDraw behavior can be further controlled through the use of negative prompts, such as "a messy drawing", to encourage the opposite behavior.

## 2 Related Work

**Text-to-Image Synthesis.** This work greatly draws from the field of text-to-image synthesis, whose primary aim is to generate images that correctly match a given textual description. In recent years, focus has been on methods that aim to learn a direct text-to-image mapping function, often in the form of a conditional GAN (Goodfellow et al., 2014a; Mirza and Osindero, 2014; Reed et al., 2016; Frolov et al., 2021). Commonly used datasets include Oxford-120 Flowers (Nilsback and Zisserman, 2008), CUB-200 Birds (Wah et al., 2011), and COCO (Lin et al., 2014), all of which contain natural images and captions describing them. While GAN-based methods have enabled considerable progress towards photorealistic image synthesis, strong autoregressive models have achieved similar quality results (Oord et al., 2017; Chen et al., 2020), with the recent DALL-E model (Ramesh et al., 2021) showcasing the benefit of scaling text-to-image synthesis networks to a large capacity. In comparison to text-to-image generative models, which require large amounts of training, this work follows the framework of *synthesis through optimization*, in which images are generated through evaluation-time optimization against a given metric.

**Synthesis Through Optimization.** Instead of directly learning an image generation network, an alternative method of image synthesis is to optimize towards a matching image during evaluation time. This framework is often referred to as *activation maximization* (Erhan et al., 2009; Nguyen et al., 2016; Mordvintsev et al., 2015), where a random image is optimized through backpropogation to increase certain neuron activations of a pretrained network. Activation-maximization methods have produced highly realistic images, however it is a challenge to understand the meaning of a neuron activation. CLIPDraw builds off a set of methods where rather than maximizing an activation, the objective is to minimize the distance between the produced image and a given description phrase, as defined by a powerful CLIP language-image encoder (Fernando et al., 2021; Murdock, 2021;

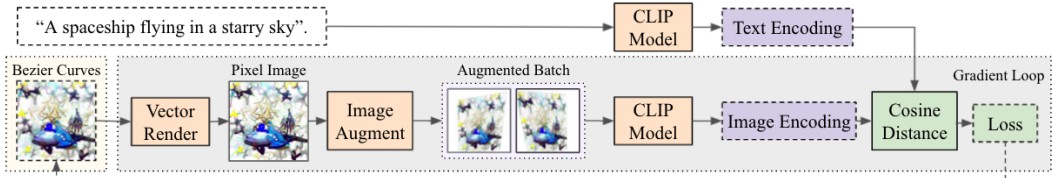

Figure 2: **CLIPDraw iteratively synthesizes images through evaluation-time gradient descent.** Starting from a random set of Bézier curves, the position and colors of the curves are optimized so that the generated drawings best match the given description prompt. Before being passed into the CLIP encoder, drawings are augmented into multiple perspective-shifted copies.

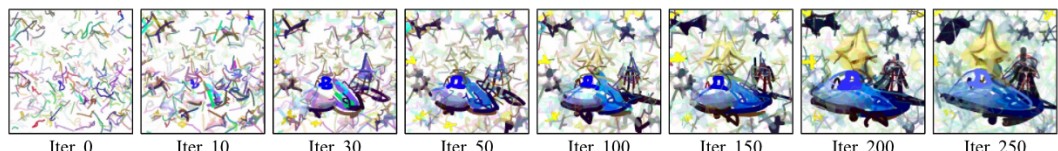

| Iter. 0 | Iter. 10 | Iter. 30 | Iter. 50 | Iter. 100 | Iter. 150 | Iter. 200 | Iter. 250 |

Figure 3: **A typical CLIPDraw run gradually forms curves into concrete shapes.** Here, a background of star-shaped structures develops into a large spaceship. More pronounced stars eventually appear, in addition to a Darth-Vader-like figure on the spaceship.

Crowson et al., 2022; Galatolo et al., 2021). A key issue in synthesis through optimization is that the produced images often leave the space of natural images (Nguyen et al., 2015), or fool the system through adversarial means (Goodfellow et al., 2014b), thus a body of work aims to discover 'natural image priors' to constrain produced images (Nguyen et al., 2016, 2017). While a typical solution is to constrain optimization to the generative space of a GAN, this setup can be expensive to evaluate, and only allows synthesis of images producible by the GAN generator. Because CLIPDraw focuses on synthesizing drawings rather than realistic pictures, CLIPDraw instead limits optimization to a set of vector curves. This constraint results in stroke-based images, which must capture larger features such as shapes and outlines, rather than fine-grained textures.

**Vector Graphics.** This work builds largely from work by Li et al. (2020), which introduces a differentiable renderer for vector graphics. Image generation methods that operate over vector images have traditionally required a vector-based dataset, however recent work has shown how differentiable renderers can be used to bypass this limitation (Reddy et al., 2021; Shen and Chen, 2021; Kotovenko et al., 2021). CLIPDraw uses a differentiable renderer as a representation for generating drawings; namely a set of RGBA Bézier curves are optimized rather than pixels.

**Follow Up Work to CLIPDraw.** A draft of this work was previously released to the public research community, and a number of follow-up papers have since been published. StyleCLIPDraw (Schaldenbrand et al., 2021) introduces an explicit image-based style loss to the CLIPDraw framework. CLIP-CLOP (Mirowski et al., 2022) extends CLIPDraw to use image patches rather than strokes, and CLIPasso (Vinker et al., 2022) uses CLIP to extract sketches from photos. Tian and Ha (2022) pursure optimization via evolutionary strategies rather than gradient descent. CLIPDraw has already influenced the image synthesis community in its pre-release form, and we hope a formal publication will continue to encourage collaboration.

## 3 Method

The objective of CLIPDraw is to synthesize a drawing that matches a given description prompt (Figure 1). Specifically, a pre-trained CLIP model is used as a judge. A CLIP model contains two networks – an image encoder and a textual encoder – which both map their respective inputs into a shared encoding space of a 512-length vector. Similarity is measured via the cosine distance between two encodings. Thus, the goal of CLIPDraw is to produce an image which, when encoded via CLIP, matches the CLIP encoding of the prompt.

Drawings in CLIPDraw are represented by a set of differentiable RGBA Bézier curves, following the method by Li et al. (2020). Each curve is parametrized by 3-5 control points, along with thickness

---

**Algorithm 1** CLIPDraw

**Input:** Description Phrase *desc*; Iteration Count $I$; Curve Count $N$; Augment Size $D$; Pre-trained CLIP model.
**Begin:**
Encode Description Phrase. *EncPhr = CLIP(desc)*
Initialize Curves. *Curves$_{..N}$ = RandomCurve()*
**for** $i = 0$ **to** $I$ **do**
    Render Curves to Pixels. *Pixels = DiffRender(Curves)*
    Augment the Image. *AugBatch$_{..D}$ = Augment(Pixels)*
    Encode Image. *EncImg = CLIP(AugBatch)*
    Compute Loss. *Loss = −CosineSim(EncPhr, EncImg)*
    Backprop. *Curves ← Minimize(Loss)*
**end for**

---

and an RGBA color vector. Drawings initially begin with curves randomly distributed throughout the image, with a white background. During optimization, the number of curves and control points is fixed, however the positions of the points along with the thickness and color vectors can be optimized via gradient descent.

The CLIPDraw algorithm (Algorithm 1) works by running evaluation-time gradient descent, as shown in Figure 2. First, the description phrase is encoded via the CLIP model, and a random set of $N$ Bézier curves are initialized. During each iteration, the curves are rendered to a pixel image via the differentiable renderer, and the resulting image is then duplicated $D$ times and augmented by a random perspective shift and random crop-and-resize. The resulting batch of augmented images is passed into the CLIP image encoder, and the cosine distances to the description phrase are summed to form the loss value. Because all operations are differentiable, gradient descent can be run through the entire loop, optimizing the parameters of the curves to decrease loss. This procedure is repeated $I$ times, until convergence.

The goal of the image augmentation is to force drawings to remain recognizable when viewed through various distortions. Without image augmentation, synthesis-through-optimization methods often result in adversarial images that fulfill the numerical objective but are unrecognizable to humans. This work uses the `torch.transforms.RandomPerspective` and `torch.transforms.RandomResizedCrop` functions in sequence. Note that the specific details of the augmentation were not the focus of this work, furthermore, trials with different augmentations show that augmentation choice does not influence synthesis to a noticeable degree (see Appendix).

Figure 3 showcases the gradual synthesis of a typical CLIPDraw drawing. Note that while the optimization process is largely deterministic, there is randomness in the initial curves and image augmentations, thus multiple runs of CLIPDraw can result in different drawings.

## 4 Results

In the following sections, various interesting behaviors of CLIPDraw are highlighted through a variety of examples. With the exception of Figure 15, example images are picked to best convey the behavior in consideration. Focus is placed on qualitative observations, unusual behavior, or recurring trends in CLIPDraw image synthesis.

Compared to methods that learn a direct generative model, optimization-based synthesis methods do not require prior training. Instead, images are generated through an evaluation-time optimization loop, aiming to maximize a given objective. This work specifically focuses on synthesizing images that match the CLIP encoding of a description prompt. The following methods are compared:

- **CLIPDraw**, in which drawings are produced by a set of RGBA Bézier curves. The control points, thickness, and colors of the curves can all be adjusted.

- **Pixel Optimization**, which instead optimizes a 224x224x3 matrix of RGB pixels. Otherwise, all algorithmic aspects are the same as CLIPDraw, including image augmentation.

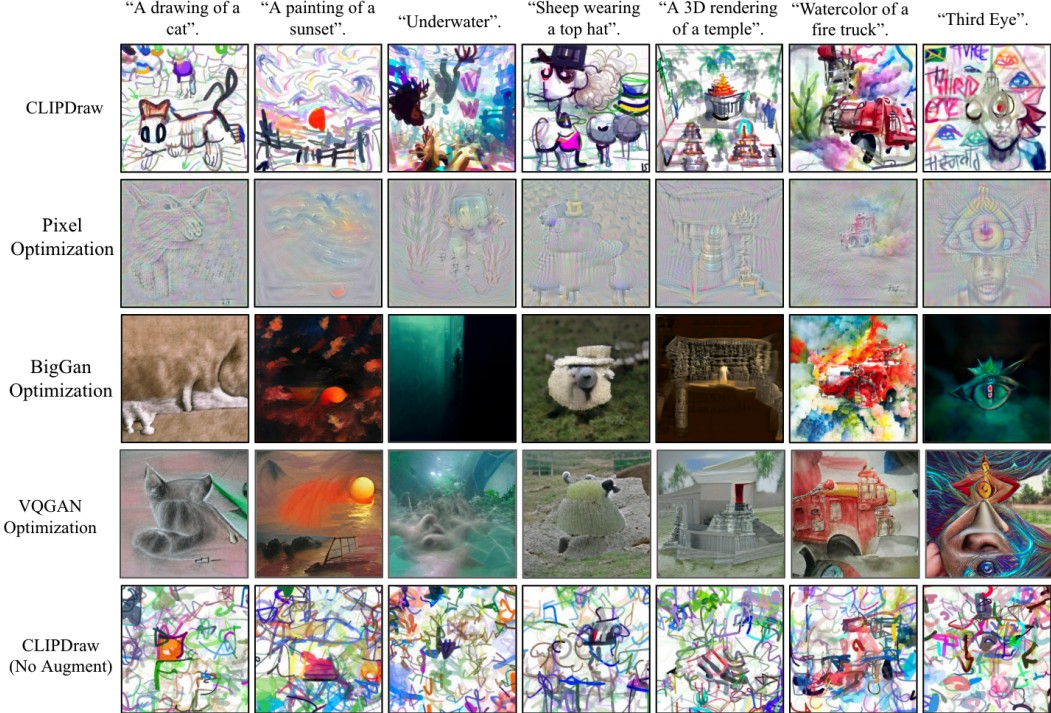

Figure 4: **Images synthesized via various synthesis-through-optimization methods**, all which share the objective of matching a given CLIP-encoded description phrase. All methods optimize for cosine similarity, as shown in Table 1. CLIPDraw can produce a diverse set of human-recognizable drawings based on simple strokes and shapes. CLIPDraw tends to result in simple drawings, often featuring multiple instances of concepts in a variety of styles. On the other hand, Pixel Optimization creates interesting textures but fails to compose colors and shapes. BigGan and VQGAN Optimization can synthesize high-resolution images, but are constrained to the set of natural images the generators are trained on, thus they tend to produce images featuring a single object, and cannot achieve techniques such as writing text. CLIPDraw without image augmentation produces images that score high during training, but fail to retain these scores when tested on augmentations, and are nonsense when viewed by humans. Cosine similarities for each prompt/method pair are shown in Table 1.

- **BigGAN Optimization**, in which images are produced using a pre-trained BigGAN generator. The weights of the generator are frozen; only the latent $Z$ vectors are optimized. Samples are generated as by Murdock (2021).
- **VQGAN Optimization**, in which images are created through sampling a VQGAN codebook (Esser et al., 2021). Samples are generated as by Crowson et al. (2022).
- **CLIPDraw (No Augment)**, which is identical to CLIPDraw, except no image augmentation is applied to the synthesized drawings.

These methods are run on the same CLIP matching objective for 250 steps of gradient descent (Figure 15). In CLIPDraw, stroke count is 256, and 8 duplicates are used during image augmentation. CLIPDraw tends to result in a diverse set of human-recognizable doodles. Pixel Optimization creates interesting textures but fails to compose colors and shapes. BigGAN and VQGAN Optimization synthesize high-resolution images, but are constrained to the set of images the generator can produce. Images from CLIPDraw without image augmentation score high numerically, but are nonsense to the human eye.

## 4.1 What kinds of visual techniques does CLIPDraw use to satisfy the textual description?

CLIPDraw often results in drawings that match their description prompts in multiple, unexpected ways, as shown in Figure 5. A prime example is the prompt for "a painting of a starry night sky". The

| Prompt | CLIPDraw | Pixel Opt. | BigGan Opt | VQGAN Opt | (No Aug) |
|---|---|---|---|---|---|
| A drawing of a cat. | .376 ± .005 | **.385 ± .009** | .325 ± .014 | .377 ± .004 | .240 ± .015 |
| A paint. of a sunset. | **.390 ± .004** | .215 ± .010 | .379 ± .003 | .379 ± .003 | .325 ± .015 |
| Underwater. | **.413 ± .006** | .385 ± .009 | .327 ± .013 | .358 ± .006 | .247 ± .004 |
| Sheep wearing a top hat. | **.434 ± .010** | **.434 ± .010** | .321 ± .016 | .411 ± .009 | .215 ± .011 |
| A 3D rend. of a temple. | **.467 ± .005** | .385 ± .009 | .311 ± .028 | .430 ± .008 | .240 ± .009 |
| Watercol. of a firetruck. | **.507 ± .007** | .385 ± .009 | .318 ± .014 | .457 ± .007 | .233 ± .007 |
| Third Eye. | **.436 ± .010** | .385 ± .009 | .323 ± .015 | .368 ± .005 | .251 ± .013 |

Table 1: **CLIP cosine similarities for images shown above.** Scores are computed by taking the average cosine similarity between the description prompt and 64 augmentations of the generated images. Each method is run for 250 steps of gradient descent, with 10 starting seeds for every prompt, and resulting means and standard deviations are presented. In CLIPDraw results, stroke count is 256, and 8 duplicates are used during image augmentation. Images take around 1-2 minutes to synthesize on a typical Colab GPU.

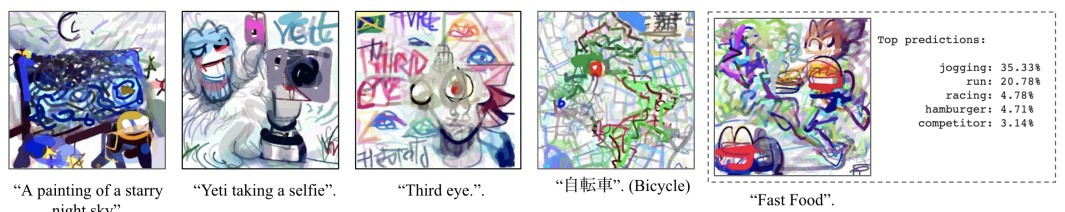

"A painting of a starry night sky".    "Yeti taking a selfie".    "Third eye.".    "自転車". (Bicycle)    "Fast Food".

Figure 5: **CLIPDraw often matches the description prompt through creative techniques**, such as forming letters inside the images, or interpreting ambiguous words in multiple ways.

drawing's background features a sky with a prominent moon and a few scattered stars. The drawing is rendered in a painterly style, however it also features an actual canvas and painter. Inside the canvas, black and blue swirls resemble Van Gogh's 1889 "The Starry Night".

Another interesting behavior of CLIPDraw is its tendency to write words in the drawing itself. In "Yeti taking a selfie", letters resembling "Yeti" can be seen in the top-right corner. In "Third Eye", again words resembling "third" and "eye" are scattered throughout the image. At times, the drawings contain symbols that are not literally the description, but are still associated, such as the prompt "自転車" (*bicycle* in Japanese) resembling a Google Maps screenshot with a Japanese-like character in the corner.

Prompt ambiguity also presents intriguing results. In "Fast Food", a McDonald's logo along with a set of hamburgers is shown. However, also present are two joggers in a footrace, providing another interpretation of "fast". Included in Figure 5 are the top words predicted by CLIP as closest to the image, showing that CLIP recognizes both "jogging" and "hamburger" as related to the synthesized drawing of "fast food".

## 4.2 Can CLIPDraw reliably produce drawings in different styles?

A useful feature of CLIPDraw is its ability to adjust not just the content of its drawings, but also the styles, based on the description prompts given. Part of this flexibility is due to the robustness of curve-based images: in comparison to methods that use a pre-trained GAN generator, CLIPDraw drawings are not limited to the space of natural images. Thus, a variety of styles can be produced, and these styles are easily explorable through text.

As shown in Figure 6, a synthesized image of a cat can look vastly different depending on the descriptor words included. When asked for a "drawing of a cat", CLIPDraw synthesized a cartoonish depiction of a cat, comprised mostly of an outline and simple face. A "realistic photograph" features more detailed shading, while a "cat as 3D rendered in Unreal Engine" showcases complex lighting along with a depth-based blurring. Further styles feature a bias towards certain colors, such as the reds and greens of Japanese woodblock prints, or the multi-color blends of watercolors.

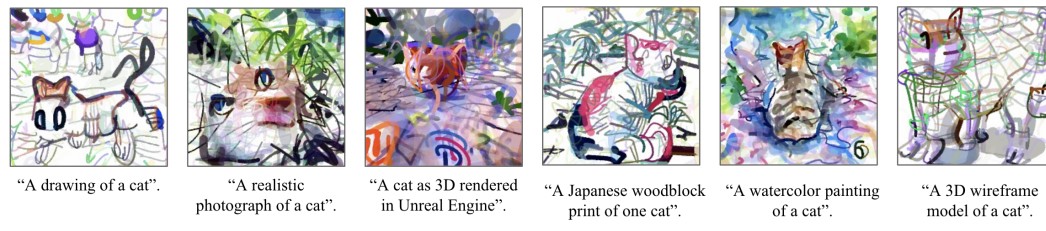

"A drawing of a cat".    "A realistic photograph of a cat".    "A cat as 3D rendered in Unreal Engine".    "A Japanese woodblock print of one cat".    "A watercolor painting of a cat".    "A 3D wireframe model of a cat".

Figure 6: **By adjusting adjectives, CLIPDraw can produce drawings of diverse styles.** Styles vary not only in the texture of the images, but showcase different representations of the underlying content, such as a cartoonish cat when prompted for a "drawing", versus a cat in perspective when prompted for a "3D wireframe model".

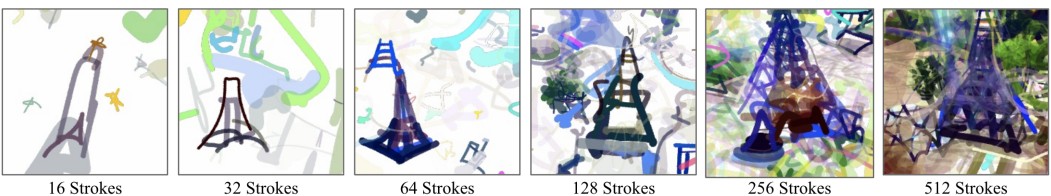

16 Strokes    32 Strokes    64 Strokes    128 Strokes    256 Strokes    512 Strokes

Figure 7: **When stroke count is increased, CLIPDraw produces drawings of increasing realism.** Low-stroke drawings of "The Eiffel Tower" opt for a cartoonish representation, while high-stroke drawings capture 3D depth, background content, and shading.

An interesting result is that adjusting descriptive adjectives not only changes the textures of the drawings, akin to Style Transfer methods (Gatys et al., 2015), but also changes its structural representation of the underlying content. For example, prompting for "a drawing" produces a flat cartoonish cat, while prompts like "a 3D wireframe" produce a cat in perspective, with depth and shadows.

### 4.3 How does the stroke count affect what drawings CLIPDraw produces?

When the stroke count is low, CLIPDraw tends to produce cartoonish or abstract representations of the given description prompt. As stroke counts increase, drawings become more detailed and incorporate additional features. Figure 7 shows "The Eiffel Tower" with various stroke counts. In the 16-stroke example, the tower is drawn as only a few straight lines. Higher stroke count images begin more details on the Eiffel Tower itself, along with additional features such as background colors and complex lighting.

A common thread in synthesis-through-optimization methods is that pure optimization leads to undesirable results; it is also necessary to constrain optimization to a suitable space of images, such as the natural images generated by a GAN, or any image made of strokes in the case of CLIPDraw. Limiting stroke count furthers this constraint. When optimizing within the space of 16-stroke images, it is hard to achieve details or textures, thus synthesized drawings will reveal the most basic forms that make up a visual concept. As a tool for AI-assisted art, the stroke-count parameter presents an easy way of adjusting between "simple" and "complex".

### 4.4 What happens if abstract words are given as a description prompt?

When given an abstract prompt without a literal interpretation, CLIPDraw must utilize cultural connections to come up with visual concepts that relate to the description. Often, this results in drawings that contain symbols relating to the given prompt, such as in Figure 8 with "Happiness" containing smiling faces and fireworks, "Translation" showcasing English and Japanese-like characters, and "Enlightenment" featuring a prominent monk-like figure.

At times, synthesized drawings demonstrate concepts through more complex relationships. In the prompt "Self", the resulting drawing features a body with multiple heads, evoking e.g. the idea that a person's self may contain multiple outward personalities. When asked "What do you look like, CLIPDraw?", the synthesized drawing contains a smiling face followed by text resembling

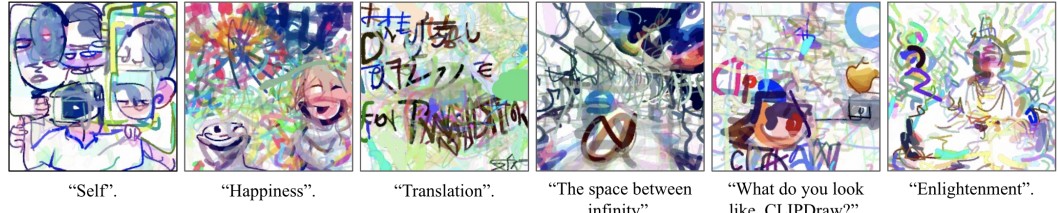

| "Self". | "Happiness". | "Translation". | "The space between infinity". | "What do you look like, CLIPDraw?". | "Enlightenment". |

Figure 8: **Abstract prompts grant insight into how CLIP relates visual concepts.** Synthesized images often contain symbols that indirectly relate to the description prompt through a cultural connection.

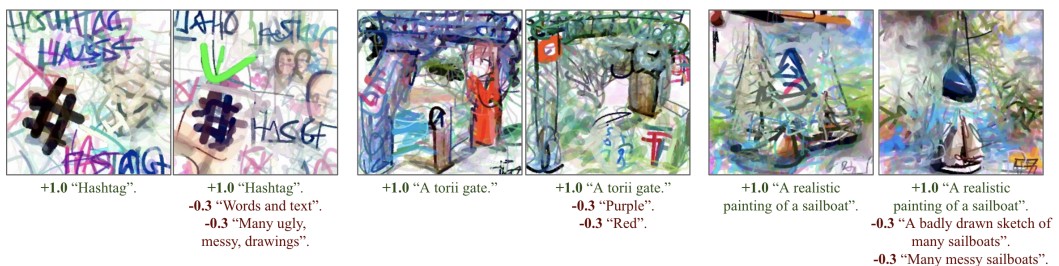

| **+1.0** "Hashtag". | **+1.0** "Hashtag". **-0.3** "Words and text". **-0.3** "Many ugly, messy, drawings". | **+1.0** "A torii gate." | **+1.0** "A torii gate." **-0.3** "Purple". **-0.3** "Red". | **+1.0** "A realistic painting of a sailboat". | **+1.0** "A realistic painting of a sailboat". **-0.3** "A badly drawn sketch of many sailboats". **-0.3** "Many messy sailboats". |

Figure 9: **CLIPDraw behavior adjusted through negative description prompts.** Negative prompts discourage synthesized drawings from matching with them, presenting a tool for fine-tuning.

"CLIPDRAW". Finally, "The space between infinity" presents a dream-like landscape with an infinity symbol under a galaxy-filled sky.

### 4.5 Can synthesized drawings be fine-tuned via additional negative prompts?

A common pain point in AI-assisted image synthesis is that is hard to control what the AI will produce. One potential tool for increased control in CLIP-based methods is to introduce negative prompts; the optimization objective is to minimize cosine distance between the CLIP-encoded drawing and the description prompt, while maximizing distance between the drawing and a set of negative prompts.

Figure 9 shows pairs of drawings synthesized from the same random initialization, with the bottom row utilizing additional negative fine-tuning prompts, weighted .3:1. "Hashtag" contains many instances of the word "hashtag" written out. By penalizing "Words and text", the bottom example contains fewer words. By penalizing "Purple" and "Red", the main color of "A torii gate" switches to green. Lastly, the original drawing for "a realistic painting of a sailboat" features many sailboats on an ocean, and penalizing the phrase "many sailboats" results in only one sailboat.

While negative prompts present a richly semantic way to fine-tune image synthesis in CLIP-based methods, it remains tricky to locate prompts that consistently encourage the intended behavior. Many times, negative prompts show negligible effects. During experiments, a cure-all negative prompt such as "a low-quality drawing", with the goal of consistently improving drawing quality, was unable to be found. Further work remains on how to best influence CLIP-based synthesis-through-optimization methods through additional objectives, whether negative or positive.

## 5   Discussion

This work presents CLIPDraw, a text-to-drawing synthesis method based on the CLIP language-image encoder. CLIPDraw does not require any model training; rather, drawings are synthesized through iterative optimization during evaluation time. CLIPDraw is not the first method to use evaluation-time optimization for image synthesis; many recent works have also used CLIP as an objective. However, by constraining synthesis to images made of RGBA Bézier curves, CLIPDraw biases towards sketches of human-recognizable concepts.

At a higher level, CLIPDraw is work that takes advantage of an existing large pre-trained foundation model (Bommasani et al., 2021), and derives benefit from the representations provided. As foundation models are often viewed as black-box models, it is important to study their use and affect on downstream tasks. Thus, the focus of this paper is to examine the nuances of CLIPDraw behavior, and experiments focus on specific questions and observations about synthesized drawings. These observations, such as prompt ambiguity or the compositionality of styles and content, are qualitatively distinct and interesting and likely apply to further CLIP-based work in the future, thus this paper aims to start an investigation that continues forward.

## 5.1 Limitations

Synthesizing high-resolution images is a challenge, and CLIPDraw will often fall short of methods that incorporate a high-functioning generative model. This problem is related to a classic pitfall in synthesis-through-optimization methods, which is that an image may very closely match the CLIP objective, while looking messy and ugly to a human. Thus it is important to introduce auxiliary objectives or constraints. In the case of CLIPDraw, drawings are constrained by the Bézier curve representation, however stricter constraints such as fooling a GAN discriminator may improve quality. However, this work shows that simple image augmentation is enough to unlock the robustness of CLIP representation to generate recognizable images.

A second limitation has to do with using CLIP encodings as a synthesis objective. While CLIP provides a rich textual representation for describing an image, in comparison to coarser neuron-activation objectives, it still remains a challenge to specify details. For example, it is hard to tell CLIPDraw to move a sailboat to the other side of the image. Preliminary experiments also explored negative prompts as a possible direction towards more fine-grained adjustments, however a consistently satisfying method was hard to locate. A promising path in future research can lie in how to correctly steer synthesized images, or introduce finer detail into description prompts via additional objectives.

## 5.2 Ethics and Social Biases

An important concept to keep in mind when dealing with human data is the existence of inherent social biases contained within. The pre-trained CLIP model is trained on a large corpus of online data, so its representations may include connections or biases that are undesirable. As CLIPDraw does not learn a new model, but instead optimizes based on CLIP itself, the bias studies presented in the CLIP paper (Radford et al., 2021) are highly relevant for CLIPDraw as well. In Section 4.4, a use case for CLIPDraw is mentioned as a tool for exploring visual connections in human culture. It is crucial to recognize that symbols or connections that are formed by CLIPDraw are not necessarily reflective of human culture, but rather are artifacts of the data used to train the original CLIP model. Thus, while CLIPDraw can be used to synthesize drawings that utilize cultural connections to evoke emotions or abstract concepts, it remains the duty of the user to ensure that the final product is up to desired standards.

## 5.3 Conclusion

Overall, the aim of this work is to introduce CLIPDraw as an easily accessible starting point to experiment with natural language image synthesis. Due to its focus on drawings rather than photorealistic rendering, CLIPDraw presents a straightforward method to examine language-image relationships without the overhead of realism. The presented CLIPDraw implementation can generally synthesize images within a minute on a typical Colab GPU.

This paper describes various interesting behaviors of CLIPDraw, however it is not exhaustive. CLIP-based text-to-image synthesis remains a field with many promising directions, and we hope others will continue to use this work for additional research into the nuances of synthesis-through-optimization, or as a practical tool for AI-assisted art and other interactive visual applications. To this end, source code is available at: https://colab.research.google.com/github/kvfrans/clipdraw/blob/main/clipdraw.ipynb.

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
