# OpenReview forum: "CLIPDraw: Exploring Text-to-Drawing Synthesis through Language-Image Encoders"
_NeurIPS.cc/2022/Conference — NeurIPS 2022 Accept_

### Official Review · Reviewer_2QeJ · 2022-07-10

**Rating:** 7
**Confidence:** 3
**Soundness:** 3 good
**Presentation:** 4 excellent
**Contribution:** 3 good

**Summary:**

The paper introduces a text-based image synthesis method by leveraging the CLIP model. The proposed method, CLIPDraw, directly optimizes a set of vector strokes without training a separate model. More specifically, control points of randomly initialized Bézier curves, thickness, color, and alpha values are fitted to match the text prompt. The paper presents a set of qualitative analyses that focus on providing insights about CLIPDraw as well as a comparison with various optimization-based approaches.

**Questions:**

1. Have the authors tried optimizing the position of the strokes? This could yield more accurate drawings with fewer strokes and hence less clutter. It could be applied together with the stroke-by-stroke approach (Fig. 11) as the stroke position might be an important degree of freedom.

**Limitations:**

Yes, a detailed and helpful discussion was provided.

**Strengths And Weaknesses:**

It is a well-written paper. The proposed approach is easy to understand. The idea of optimizing parametric curves rather than the pixels is interesting. Most importantly, it works and enables the synthesis of stylistic images without training a separate generator network. CLIPDraw achieves promising results compared to the other synthesis-through-optimization baselines. The augmentation technique also seems to be highly effective.

CLIPDraw applies perspective and resized crop augmentations that have an emphasis on the shape, which are suitable for parametric curves. If the pixel baseline (i.e., Pixel Optimization in Section 4) applies the same, I would also suggest trying a color-based augmentation. The baseline seems to work well with the shape but lacks the texture.

Ablation of the number of augmentations and the range of augmentation parameters could be useful for the reader.

This is already mentioned as a limitation in the paper. But I would like to express my interest in adding a discussion about potential solutions. The proposed method always generates images in a watercolor painting style due to the parametric curve inputs. It requires a large number of strokes to make the content recognizable. However, then it gets cluttered and without sharp details. A discussion on how to increase the quality could be insightful.

---

### Official Review · Reviewer_6hUh · 2022-07-11

**Rating:** 6
**Confidence:** 4
**Soundness:** 3 good
**Presentation:** 3 good
**Contribution:** 3 good

**Summary:**

This paper proposes an interesting CLIPDraw model that can synthesizes novel drawings from natural language input. Specifically, the proposed model follows the synthesis-through-optimization paradigm and utilizes the powerful pre-trained CLIP language-image encoder as a metric for maximizing similarity between the given description and a generated drawing. It also adopts the differentiable renderer to operates over vector strokes rather than pixel images. In addition, the rendered image are augmented to force drawings to remain recognizable when viewed through various distortions. The images synthesized by CLIPDraw looks creative and have human-recognizable semantics. The pre-release work has inspired some follow-up methods in the image synthesis community.

**Questions:**

See weaknesses above.

**Limitations:**

The authors have adequately addressed the limitations and potential negative societal impact of their work.

**Strengths And Weaknesses:**

Strengths:

+ The task and the idea are interesting.
+ The experiments verify the creativity, semantics and controllability of the proposed method in image synthesis results.
+ The work may enlighten many follow-up studies.

Weaknesses:

+ We may call the proposed model a simple yet effective model, but its effect largely depends on the existing two models: differentiable renderer (Li et al., Differentiable vector graphics rasterization for editing and learning, ACM TOG, 2020) and CLIP (Radford et al., Learning transferable visual models from natural language supervision, ICML, 2021), which makes me concern about the technical contribution of the proposed method.
+ In addition, some works also adopt techniques similar to the proposed model, which are not mentioned in this paper, such as: Parameterized Brushstrokes (Kotovenko et al., Rethinking Style Transfer: From Pixels to Parameterized Brushstrokes, CVPR, 2021) and CLIPasso (Vinker et al., CLIPasso: Semantically-Aware Object Sketching, arXiv preprint arXiv:2202.05822, 2022). The tasks of these methods may be different, but they are also used the differentiable renderer and the CLIP model, and follows the synthesis-through-optimization paradigm. Therefore, the authors should further discuss the differences and the originality of the proposed method compared with other methods.

---

### Official Review · Reviewer_i8mj · 2022-07-12

**Rating:** 7
**Confidence:** 5
**Ethics Flag:** Yes
**Soundness:** 3 good
**Presentation:** 4 excellent
**Contribution:** 3 good

**Summary:**

This paper presents a synthesis-through-optimization approach that generates drawing based on textual description. The approach leverages differentiable rendering to optimize RGBA Bézier curves to maximize the CLIP representation between generated images and the text input. The proposed approach initialize a predefined number of curves at random positions and modify the curve parameters (color, transparency, thickness, etc.) via gradient descent.

The authors demonstrated that the proposed methods can generate visually interesting image by showing qualitative examples and demonstrate the importance of using data augmentation during synthesis optimization. In addition, the author showed that practitioners can gain more control over this method by using negative controls (additional input text that describes the visual aspect that should not appear in the generated images).

I find the proposed method to be a simple framework that can be used to visualize the knowledge captured in CLIP. Given its simplicity and effectiveness, I am leaning toward accept.

However, given that more powerful (pixel-based) image generation systems are getting developed and released, I encourage the authors to discuss more about the future aspect of this current work. Will the proposed technique still be useful in three years? For this, I have more detailed comments in the following sections.

**Questions:**

Question:
1. what is the wall clock time of the proposed method for generating one image?

Comments on Writing:

1. `Rather than photorealistic images, CLIPDraw aims to synthesize simple drawings that nevertheless match the prompt.` Why are we interested in this task?

1. `Thus, CLIPDraw optimizes a set of vector strokes rather than pixel images, a constraint that biases drawings towards simple human-recognizable shapes.` Argubly, methods that generate pixel images are also human interpretable. I don’t quite see this shortcoming of recent works in this regard.

**Limitations:**

I do not see any negative societal impact of this work

**Strengths And Weaknesses:**

Strength:
1. the proposed method is novel
2. the proposed method is simple and easy to implement
3. the proposed method has already generated impact in this direction of work

Weakness:
1. Given that pixel-based image generation methods are becoming increasingly powerful, I am not sure if the proposed method would still stand out.
2. most of the evaluation is done by example visualizations, which makes it hard to assess the general quality of the proposed method

---

### Author Response · Authors · 2022-08-02
**Author Reply**

Thank you all for the valuable reviews. Since the feedback was largely positive, we will not be making any large changes to the work.
However, some comments will be addressed in the revision:
- Wall clock time is ~1 minute on a Colab GPU
- The motivation for focusing on human-interpretable strokes (in contrast to photorealistic images) lies in the simplicity and flexibility of the objective. Photorealistic objectives work well, however they have been the subject of many works, and there is a benefit in studying an image basis which focuses on higher-level visual representation rather than pixel-level style and details.
- We now cite Parametrized Brushtrokes and CLIPPasso
- Ablations on the image augmentation method are included in the Appendix

---

### Meta-Review · Area_Chair_p5LC · 2022-08-29

**Recommendation:** Accept
**Confidence:** Certain

**Metareview:**

This is a very interesting paper. While there are methods for generating text without training using CLIP (e.g., https://arxiv.org/abs/2205.02655), this paper introduces a method generating stroke-based images based on the similarity between the text and the image. The performance of the method is quite impressive and the reviews are all positive. I therefore recommend acceptance of this paper.

**Award:**

No

---

### Decision · Program_Chairs · 2022-09-14

Accept